# Reconfigurable emergent patterns in active chiral fluids

Bo Zhang [1], Andrey Sokolov [1] & Alexey Snezhko [1✉]

Active fluids comprised of autonomous spinning units injecting energy and angular momentum at the microscopic level represent a promising platform for active materials design. The complexity of the accessible dynamic states is expected to dramatically increase in the case of chiral active units. Here, we use shape anisotropy of colloidal particles to introduce chiral rollers with activity-controlled curvatures of their trajectories and spontaneous handedness of their motion. By controlling activity through variations of the energizing electric field, we reveal emergent dynamic phases, ranging from a gas of spinners to aster-like vortices and rotating flocks, with either polar or nematic alignment of the particles. We demonstrate control and reversibility of these dynamic states by activity. Our findings provide insights into the onset of spatial and temporal coherence in a broad class of active chiral systems, both living and synthetic, and hint at design pathways for active materials based on self-organization and reconfigurability.

---

[1] Materials Science Division, Argonne National Laboratory, 9700 South Cass Avenue, Argonne, IL 60439, USA. ✉email: snezhko@anl.gov

Assemblies of motile particles are prime examples of active materials that exhibit fascinating collective behavior and form dynamic self-organized structures away from equilibrium[1–4]. Shape anisotropy plays an essential role in the development of a collective motion and self-organization in both biological and synthetic active systems at multitude of length-scales from animal flocks and collections of robots to bacterial suspensions, molecular-motors filamentary proteins mixtures, and self-propelled colloidal particles[5–17]. It is often a major factor that defines a direction of self-propulsion, gives rise to a symmetry breaking in ordered phases, and promotes a polar or nematic order in active matter systems[18,19]. Besides ensembles with linearly translating active particles, there is now a strong interest in systems comprised of chiral active units capable of changing the direction of their motion autonomously or in response to external stimuli[20]. Variety of microorganisms such as bacteria[21,22] and sperm cells[23], magnetotactic bacteria in rotational magnetic fields[24], microtubules-molecular-motors mixtures[25] represent realizations of chiral active matter. Synthetic circle swimmers have been realized by a design of L-shaped self-phoretic particles[26] or actuated by rotational magnetic filed electrophoretically driven Janus colloids[17].

Active spinner fluids, comprised of autonomous spinning units, enable unique dynamic phases and transport characteristics[27–31]. A significant effort, both experimental and theoretical, has been recently dedicated to understanding of fundamental rules governing the emergence of collective states in active colloids comprised of spinning units[28,32–39]. Two complementary active systems composed of spontaneously rotating colloids use electric[27] or magnetic[40] fields to power the system. Quincke rollers[41] make use of spontaneous electro-rotation of a dielectric sphere submerged in a conducive fluid and exposed to a static electric field. In contrast, magnetic rollers rely on a spontaneous rotation of a ferromagnetic sphere in a presence of a uniaxial alternating magnetic field[40]. Both systems demonstrate a remarkable level of complex collective behavior ranging from emergence of flocks to formation of global vortices controlled by activity[38,40,42]. However, aside from systems with prescribed direction of motility[36,43] our understanding of the role shape anisotropy plays in the emergence of a collective motion and self-assembly in active spinner fluids remains unexplored. In this letter, we employ shape-anisotropic Quincke rollers to introduce active chiral rollers and reveal a plethora of novel collective dynamic patterns emerging away from equilibrium. We demonstrate how activity and anisotropy of the spinning units can be used to orchestrate collective behavior and dynamic self-organization in active chiral liquids.

## Results

**Shape-anisotropic Quincke rollers**. Our system is comprised of colloidal pear-shaped dielectric particles confined inside of a cylindrical cell and powered by a static (DC) electric field applied perpendicular to the bottom surface of the cell (Fig. 1a). Once particles sediment on the bottom of the cell and a homogeneous DC field is applied above a certain threshold, they start to spontaneously rotate due to electrohydrodynamic Quincke rotation phenomenon[41,44] and turn into rollers. The speed, $|v|$, of the rollers increases with the field strength, $E$, however it does not obey the $|v| \sim \sqrt{(E/E_c)^2 - 1}$ relation observed for spherical particles[27]. $E_c$ is a threshold electric field strength for particles rolling. Instead, it exhibits a crossover behavior at certain magnitude of the electric field (see Fig. 1b) indicative of two distinctive regimes of rolling. The shape anisotropy of the particles is responsible for the observed behavior. The growth of the particle activity triggered by an increase in the driving field strength, $E$,

results in changes of particle rolling mechanics. In contrast to the spherical rollers randomly selecting axis of the initial rotation in the plane normal to the external electric field, pear-shaped rollers favor rotations around the long axis due to a viscous drag anisotropy. As a result, pear-shaped rollers propel along the bottom of the cell with the long-axis orthogonal to the direction of motion (see Fig. 1c). Furthermore, since a pear-shaped geometry is also asymmetric along the long axis, the resulting rolling trajectories are curved. Consequently, two flavors of Quincke rollers are spontaneously realized: rollers moving on trajectories with a clockwise (CW) winding and those with counterclockwise (CCW) motion. Both chiral states (CW or CCW winding motion) of rollers are equally probable and simultaneously present in the system. Such chiral Quincke roller is a realization of a 'circle roller' in analogy with 'circle swimmers'[17,45].

We define the orientation of a pear-shaped particle as a unit vector, **n**, along the long-axis pointing from the center of the larger sphere forming the particle (body) toward the center of the smaller sphere (head). In general, the long axis of a pear-shaped roller may have a tilt, $\theta$, with respect to the plane parallel to the bottom surface of the container, and **n** can therefore point either toward the bottom surface ($\theta < 0$) or away from it ($\theta > 0$), see Fig. 1c. Experiments reveal that the tilt angle strongly depends on the activity in the system. At low activity levels corresponding to low field strengths (below 1.96 V μm$^{-1}$) and low particle velocities, the axis is tilted such that the particle orientation vector points away from the substrate ($\theta > 0$, mode $\alpha$). As the activity increases with the applied field, the tilt decreases, passes through zero ($\theta = 0$, mode $\beta$) and eventually becomes negative ($\theta < 0$, mode $\gamma$). Variations in a tilt angle of the rollers with the activity inevitably affect curvature of the corresponding trajectories and could be conveniently characterized experimentally by a normalized trajectory curvature parameter defined for nonzero tilt angles as, $\kappa = \langle (\mathbf{n}_{xy,i} \cdot \Delta \mathbf{r}_i) / (|\mathbf{n}_{xy,i} \cdot \Delta \mathbf{r}_i|) \rangle_i$. Here, $\mathbf{n}_{xy,i}$ is the projection of a particle $i$ orientation vector **n** on the bottom plane, $\Delta \mathbf{r}_i$ is a short displacement (of the order of a roller size) of the particle $i$ after a finite time, $\delta t$, and $\langle \rangle_i$ is an ensemble average. Thus, for a given activity $\kappa$ can take values from $-1$ ($\alpha$ mode-'heads-out' rolling) to 1 ($\gamma$ mode-'heads-in' regime). A crossover region $\kappa \sim$ corresponds to a $\beta$ mode . The evolution of $\kappa$ with the field strength is shown in Fig. 1d (see also Supplementary Fig. 1 for $\kappa$ versus electric field strength at different rollers area fractions). The positive–negative transition region ($E \sim 1.96$ V μm$^{-1}$) for $\kappa$ coincides with the crossover region in Fig. 1c where velocity of the pear-shaped roller strongly deviates from the behavior observed for spherical rollers. Transitions from $\alpha$ to $\gamma$ modes with the activity also inevitably lead to a chiral states reversal of the individual rollers, due to changes in the sign of the local curvature. For each mode of rolling set by the electric field strength both windings (CW and CCW) are spontaneously realized in the ensemble with equal probability.

The rolling behavior of the pear-shaped rollers is further quantified by the time evolution of the mean square displacement (MSD), $\langle \Delta \mathbf{r}^2(\tau) \rangle = \langle [\mathbf{r}(t + \tau) - \mathbf{r}(t)]^2 \rangle$ and mean square angular displacement, $\langle \Delta \varphi^2(\tau) \rangle = \langle [\varphi(t + \tau) - \varphi(t)]^2 \rangle$ at low area particle number density, $\phi = 0.001$. Representative MSD curves for different modes of rolling are shown in Fig. 1e. Initially, the rollers move ballistically as $\langle \Delta \mathbf{r}^2(\tau) \rangle \sim \tau^2$ at all regimes. Rollers in the regimes close to the transition point (blue and yellow curves in Fig. 1e) have a low curvature of their trajectories compared to the rest of the curves and transition to a nearly diffusive regime at longer times as they are less localized than rollers with high curvature of the trajectories and, therefore, have higher probability for collisions with other rollers. As curvature and activity increases (red curve) characteristic oscillations of the MSD emerge indicative of the circular-like, more localized

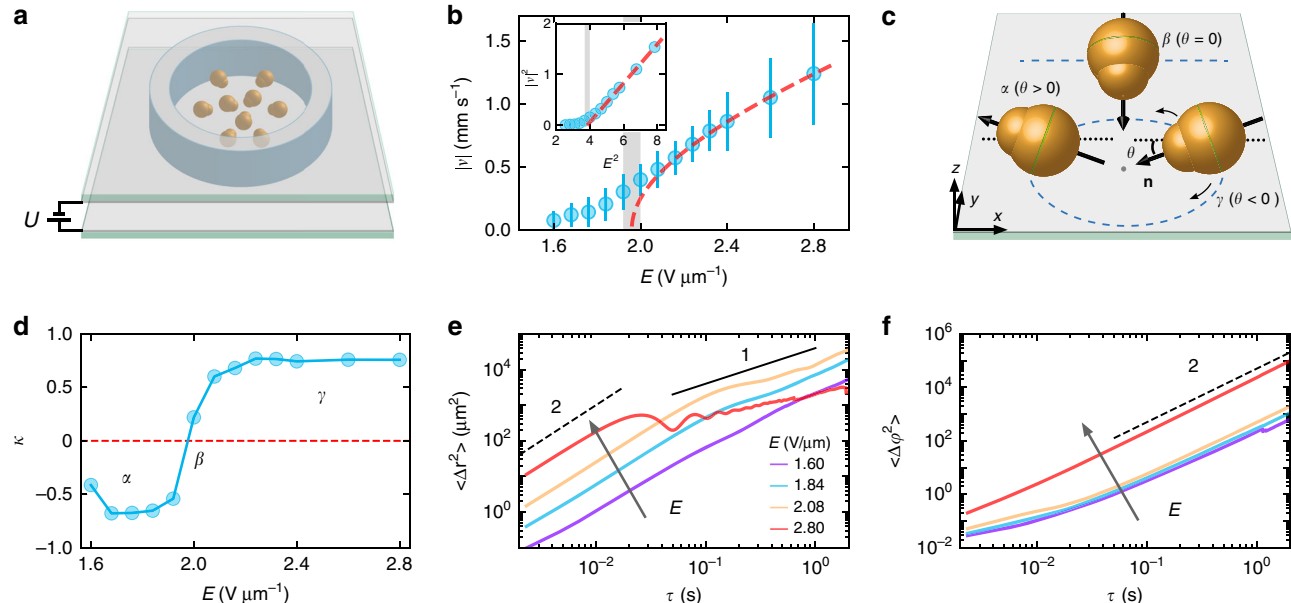

**Fig. 1 Individual particle dynamics of pear-shaped Quincke rollers. a** Schematics of the experiment: pear-shaped particles are confined in a cylindrical well. A uniform DC electric field is applied normal to the bottom surface of the cell. **b** Rolling velocity of the pear-shaped particles as a function of the electric field strength $E$. Shaded area is a crossover region between two different modes of particles rolling. The red dash line is a fit of the high field part of the curve to $|v| \sim \sqrt{(E/E_c)^2 - 1}$ dependence typical for spherical rollers. $E_c = 1.96$ V $\mu$m$^{-1}$. Insert, $|v|^2$ versus $E^2$. Error bars are standard deviation of the absolute values of velocities. **c** Three rolling modes of a pear-shaped particle: $\alpha$ ($\theta > 0$), $\beta$ ($\theta = 0$) and $\gamma$ ($\theta < 0$). Blue dash lines illustrate corresponding trajectories. **d** Trajectory curvature parameter, $\kappa$, as a function of the electric field strength. Two distinctive modes of rolling correspond to $\alpha$ mode-'heads-out' ($\kappa < 0$) and $\gamma$ mode-'heads-in' ($\kappa > 0$) regimes of particles propulsion. **e** Time evolution of the mean square displacement of pear-shaped Quincke rollers shown for a set of driving electric field strengths. **f** Mean square angular displacements of pear-shaped rollers. In **e**–**f** black solid line has a slope of 1 and black dash lines have slopes of 2. In **b**, **e**–**f** area fractions of rollers are 0.001.

motion of the rollers and as a result sub-diffusive behavior. Similar behavior is eventually observed also for $\alpha$—mode rollers (purple curve) with high curvature of the trajectories, albeit at much longer time scales due to significantly lower velocity of the rollers motion at low field strengths. The oscillations are also reflected in the variance of the in-plane particle orientations $\langle \Delta \varphi^2 (\tau) \rangle$, shown in Fig. 1f, where all regimes develop $\tau^2$ dependence indicative of predominantly circular-like motion of the rollers.

**Emergent collective motion of pear-shaped rollers**. Ensembles of pear-shaped rollers demonstrate a strong propensity toward development of a large-scale collective motion. A chiral motion of pear-shaped anisotropic rollers that could be orchestrated by the activity makes possible collective dynamic states not accessible to spherical rollers. All phases are dynamic by nature and are facilitated by a fine interplay between activity and chirality. Two of the unique discovered phases, self-organized multi-vortical patterns and synchronized rotating flocks, are shown in Fig. 2, see also Supplementary Movies 1–6. The vortices formed by pear-shaped rollers have a certain characteristic size that depends on the activity and come in two equally probable flavors: CW and CCW (Fig. 2a, b and Supplementary Movie 3). Once formed, dynamic vortices are stable, while energy is provided to the system by the electric field. Positions of vortices are not fixed and change from one experiment to another. Most rollers move around corresponding vortical centers and make circular trajectories but some at the edges of vortices may travel among multiple vortices (Fig. 2c). Vortices of the same chirality are often neighbored by vortices of the opposite chirality flavor, indicating the tendency of the system toward a 2D antiferromagnetic order (Fig. 2b). The pattern is reminiscent of self-organized vortices of both chiralities emerging in dense ensembles of collectively

moving microtubules propelled by surface-bound molecular motors[25]. There, however formation of vortices was driven by a reptation-like motion of microtubules in combination with local nematic alinement interactions due to collisions, while vortices in our system rely on long-range hydrodynamic aligning interactions discussed below.

As the activity increases driven by the growth of $E$, a different type of a flocking state, rotating flocks, emerges (Fig. 2d and Supplementary Movie 4). The rotating flocks are formed by individual chiral rollers with both rotating frequencies and particle orientations synchronized within each flock. Inside a rotating flock the orientations of all rollers follow the same polar orientation that itself rotates with time (see Figs. 3i, 4e and Supplementary Movie 5). The system usually develops multiple flocks that can spontaneously select CW or CCW sense of rotation. The velocity and vorticity fields clearly reveal stable flocking domains and domain boundaries (Fig. 2e and Supplementary Movie 6). The polarization within the rotating flocks changes with the electric field strength. It first growth with the field strength as we move away from the phase boundary with the vortex phase and then decreases due to growing localization of the chiral rollers that leads to a decreased number of interactions with other chiral rollers when approaching a phase boundary with the spinner phase (see Supplementary Note 1 and Supplementary Fig. 2). As evident from our data, the system of pear-shaped rollers exhibits a spontaneous chirality induced phase separation from initially random distribution of chiral rollers. Observed rotating flocks with spontaneous segregation of CW and CCW rotations are reminiscent of rotating droplet patterns realized in computational studies of Vicsek-like models for chiral swimmers with a circular motion[46–48].

Individual rollers within a flock usually keep their association with the particular flock. However, instances of the rollers

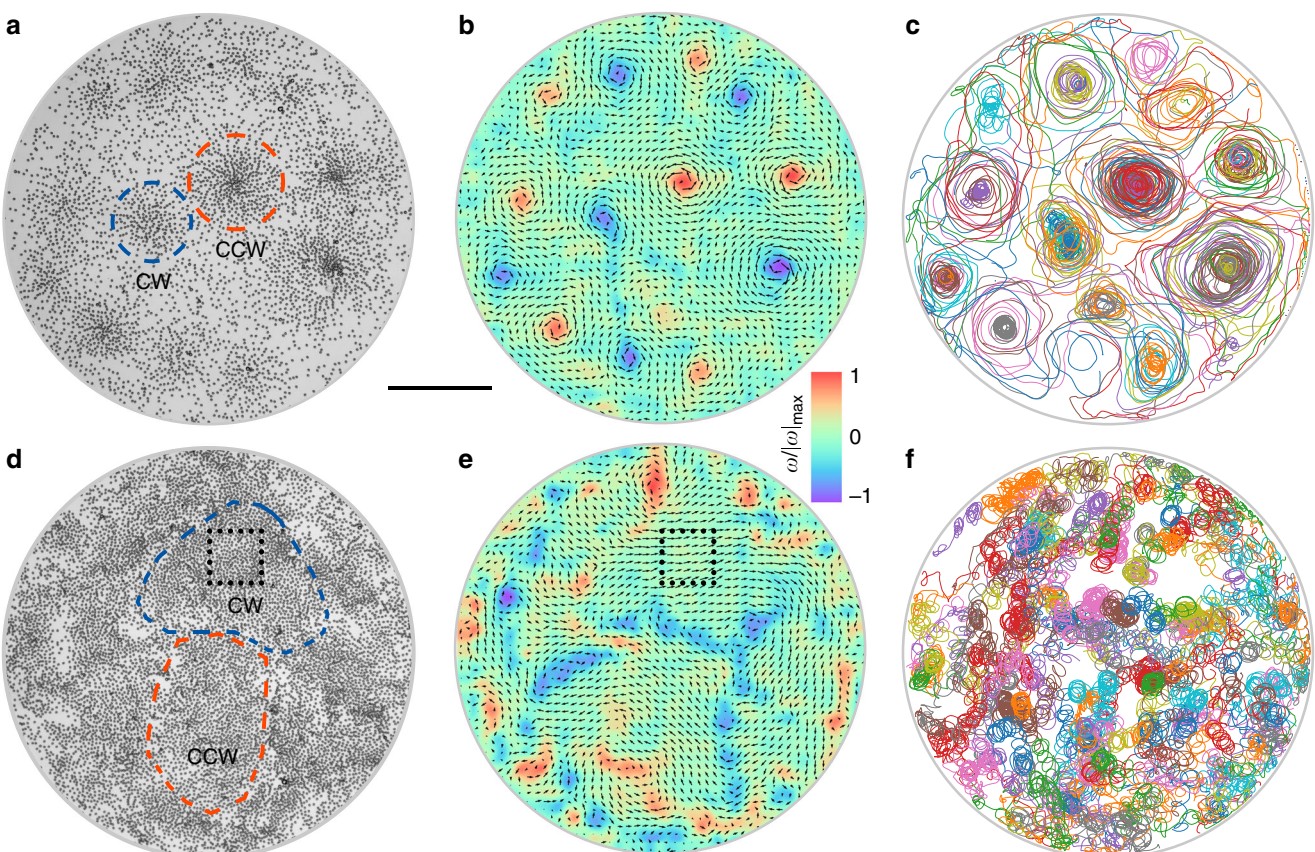

**Fig. 2 Emergent self-assembled patterns comprised of chiral rollers. a, d** Experimental image of the vortex and rotating flock phases spontaneously formed in the ensemble of active chiral rollers energized by the electric field. Both types of chiral states (CW and CCW) of vortices or rotating flocks are simultaneously present in the system. See also Supplementary Movies 1–6. **b** Superimposed velocity (arrows) and vorticity (background color) fields of the chiral rollers shown in (**a**). Quasi-antiferromagnetic order of vortex chiralities is observed. **c** Typical trajectories of the pear-shaped rollers in a vortex phase. Only 2% of particle trajectories are shown. In **a–c** particle area fraction is 0.108 and electric field strength is 1.92 V μm$^{-1}$. **e** Velocity field of the chiral rollers in rotating flocks is illustrated by arrows. The direction of particle motions are synchronized within each flock. Background color depicts a vorticity field in the rotating flocks phase. **f** Trajectories of pear-shaped rollers in a rotating flocks phase. 0.5% of the trajectories are shown for clarity. In **d–f** particle area fraction is 0.254 and electric field strength is 2.60 V μm$^{-1}$. Blue and red dashed shapes depict typical CW and CCW vortices or flocks, respectively. Black doted squares mark the position of the rotating flock region shown in detail in Fig. 3i. Scale bar is 0.5 mm.

switching the flocks are also observed mainly in the border regions of the rotating flocks (Supplementary Movie 4). Remarkably, the shapes of the flocks are often far from circular and are quite stable for a long time. No significant evolution of the rotating flocks (coarsening and merging) has been observed within a timescale of our experiments (about 10 min). It is, however, possible that the coarsening dynamics has a much larger timescale (hours or days) in this system. In the rotating flocks regime the rollers circulate and their trajectories are tendril like (Fig. 2f) and motions of particles are more localized. On the contrary, in traditional flocks such as schools of fish or bacterial suspensions the particles dynamics become super-diffusive[18].

**Phase diagram of emergent states**. A full phase diagram of different emergent states in active chiral roller system is summarized in Fig. 3a. At low activity levels, corresponding to small field strengths $E$, the state is gas like and the motion of rollers is mostly uncorrelated with intermittent clusters forming at high particle densities. Similar in nature uncorrelated gas state is observed at low particle area fraction above the curvature transition region ($E \sim 1.96$ V μm$^{-1}$). We discriminate rollers in the gas phase with the $\gamma$ mode of motion (above 1.96 V μm$^{-1}$) and call them spinners as their trajectories become more and more localized with the activity—the radius of curvature decreases and

becomes in extreme cases of the order of the particle size. As the rollers density and the field magnitude increase the vortex states emerge that in turn transition into the rotating flocks regime at high activity levels. The transition region between rotating flocks and spinners (gas of spinners) is smooth as flocks gradually shrink and fall apart with the decrease in the rollers density at fixed activity. To discriminate between these two phases in the phase diagram, we involve the temporal correlation functions of the velocity field, $C_{T,vf}$, see 'Methods' for the details and Fig. 5.

Formation of collective phases in a system of chiral active rollers is driven by velocity alignment interactions. The alignment mechanism due to inelastic particle collisions has been discussed previously in detail in several publications[49–51]. Inelastic collisions between active particles lead to velocity alignment and buildup of spatial correlations. In particular, for the systems of Quincke rollers both hydrodynamic and electrostatic interactions contribute to the velocity alignment[27]. In the case of chiral rollers, the vortical patterns emerge in the transition region between two modes of rolling and correspond to a largest persistence lengths for individual rollers (low curvature of the trajectories), that facilitate the largest interaction range (of the order of the system size) and time for rollers to interact and enhance their alignment. The size of the vortices depends on localization of the chiral rollers as demonstrated by the linear relation between the vortex sizes and the persistence length shown in Fig. 3f. As the curvature

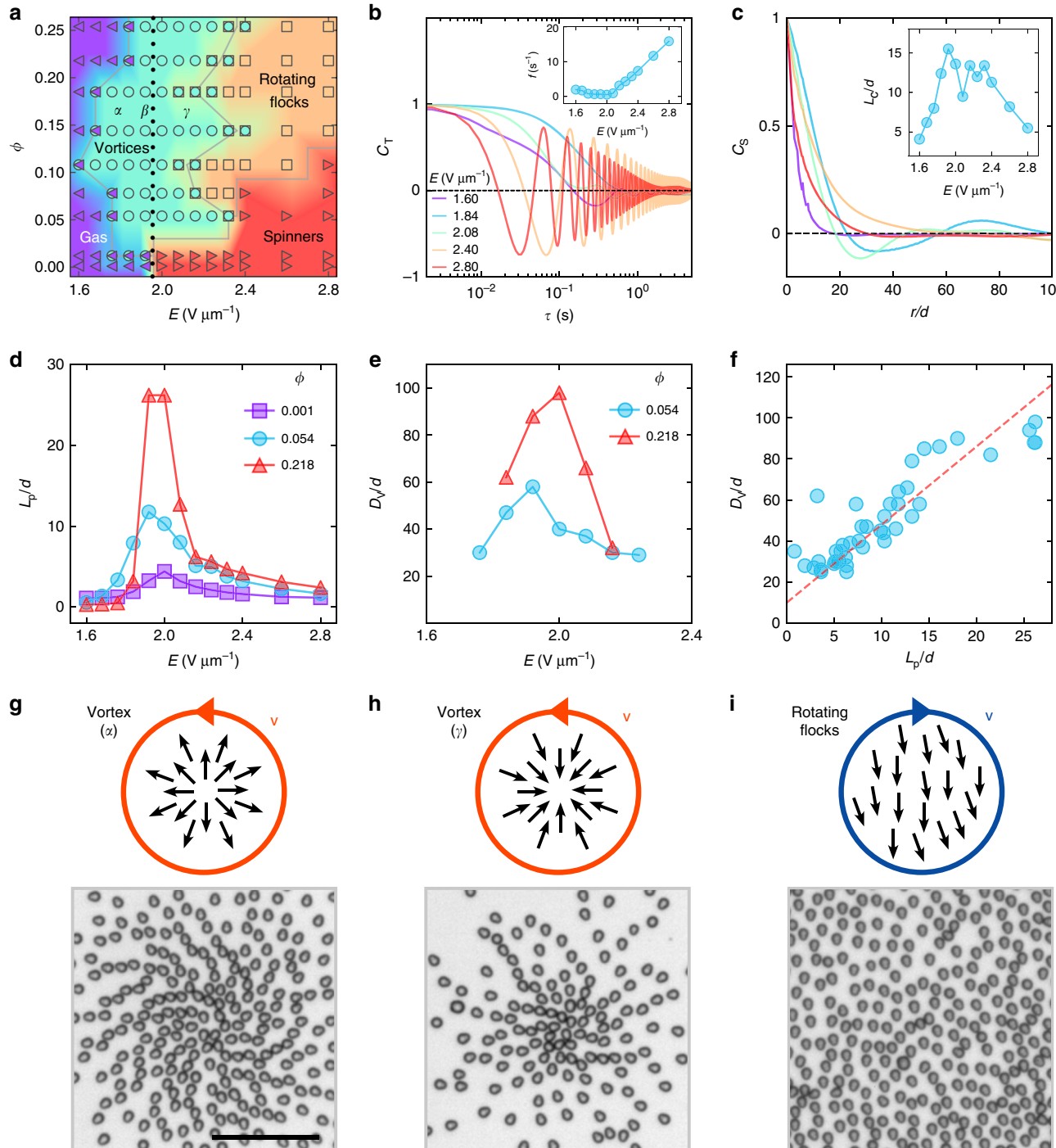

**Fig. 3 Emergent patterns in chiral rollers swarms. a** Phase diagram of the dynamic states in a chiral roller ensemble. Gas, vortices, rotating flocks and spinners phases are shown as left triangles, circles, squares and right triangles, respectively. Different phases are discriminated with the help of the temporal correlations of the corresponding velocity fields, see 'Methods'. The black dotted line indicates the transition from $\alpha$ to $\gamma$ rolling. **b** Time evolution of the velocity temporal correlation function $C_T$ of rollers shown for different magnitudes of the electric field. Insert: frequency of the rotations as a function of the electric field obtained by the fast Fourier transform (FFT) of $C_T$. $\phi = 0.108$. **c** Velocity spatial correlation function $C_s$ of the rollers at different activity levels. $d$ is the particle size. The color scheme is the same as in (**b**). Inset: the correlation lengths $L_c$ versus electric field. The presence of a valley of suppressed $L_c$ values with a local minimum at $E = 2.08$ V $\mu m^{-1}$ corresponds to the region of vortices with strong negative correlation in $C_s$. $\phi = 0.108$. **d** Persistence length, $L_p$, as a function of field strength at different area fractions $\phi$. **e** Vortex size, $D_v$, as a function of the field strength. **f** Characteristic vortex size, $D_v$ as a function of the persistence length, $L_p$. The dash line is a linear fit. **g–i** Local ordering of rollers in self-organized emergent patterns. **g** a 'heads-out,' $\alpha$-vortex, with all the rollers oriented away from the vortex center. $E = 1.84$ V $\mu m^{-1}$ and $\phi = 0.108$. **h** a 'heads-in,' $\gamma$-vortex, formed by rollers oriented toward the vortex eye. $E = 2.08$ V $\mu m^{-1}$ and $\phi = 0.108$. **i** a rotating flock. The same rotating flock is indicated by a black square in Fig. 2d, e. $E = 2.60$ V $\mu m^{-1}$ and $\phi = 0.254$. All rollers within the flock synchronously point in one direction while the direction of the flock itself rotates. Scale bar is 0.1 mm. See also Supplementary Movies 2, 5, and 7–9.

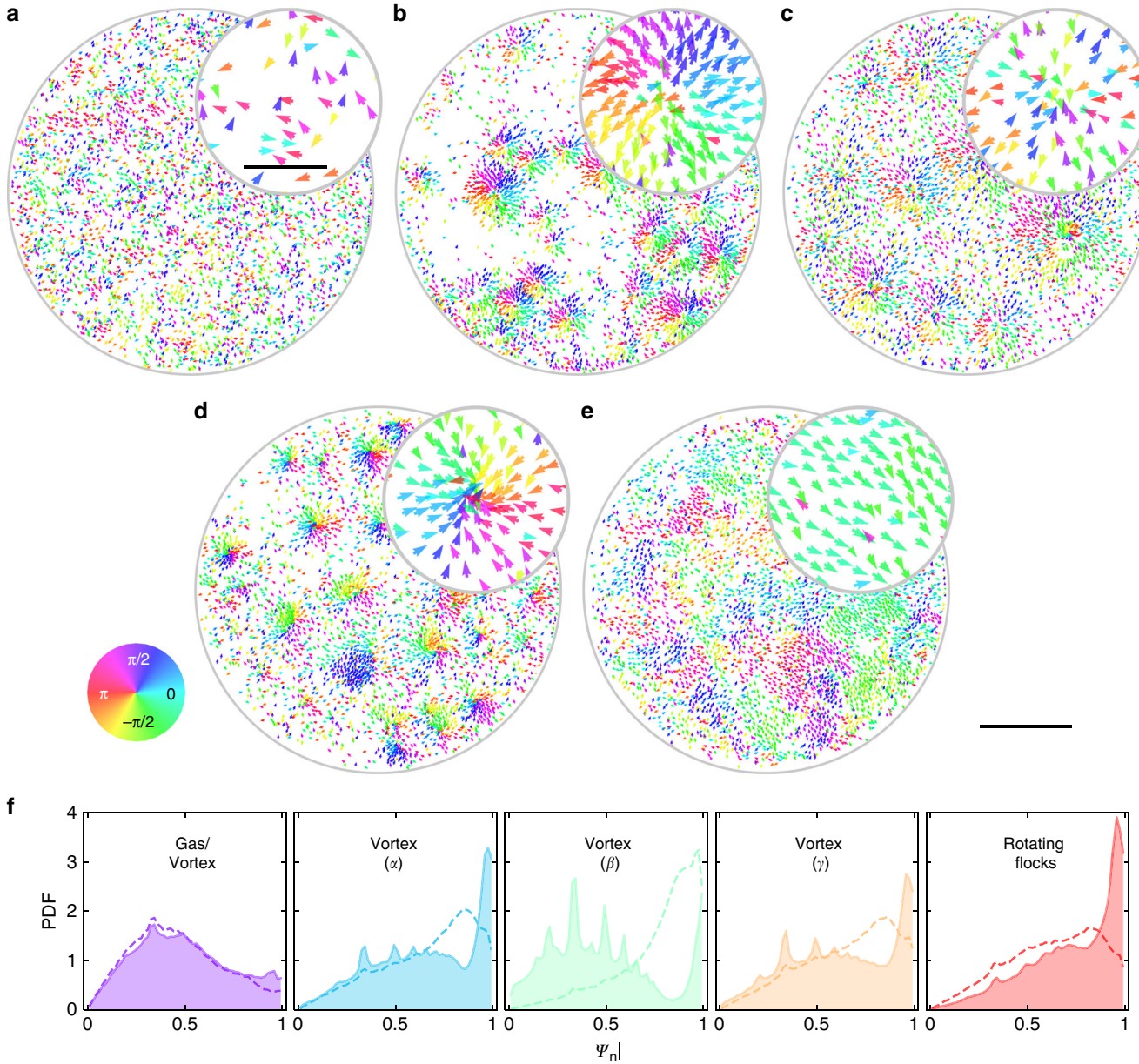

**Fig. 4 Local order inside of collective chiral rollers states. a** Uncorrelated gas of chiral rollers. $E = 1.60$ V μm$^{-1}$. **b** $\alpha$- ('heads-out') vortices. $E = 1.76$ V μm$^{-1}$. **c** $\beta$-vortices. $E = 1.92$ V μm$^{-1}$. **d** $\gamma$- ('heads-in') vortices. $E = 2.08$ V μm$^{-1}$. **e** Rotating flocks. $E = 2.40$ V μm$^{-1}$. In **a**–**e** particle orientations are shown as arrow-heads and also color coded to better visualize rotating flocks and vortex patterns in the system. Area fractions correspond to $\phi = 0.108$. The inserts demonstrate the local order and are representative zoomed-in areas from the corresponding panels. Scale bars are 0.5 mm for the main panels and 0.1 mm for the inserts. **f** Probability distribution functions of the local orientational polar order parameters, $|\Psi_1|$ (solid lines), and nematic order parameters, $|\Psi_2|$ (dashed lines), calculated for emergent collective patterns shown in (**a**)–(**e**). $\alpha$-, $\gamma$-Vortices and Rotating flocks exhibit a polar local ordering, while $\beta$-vortices show tendency toward a local nematic ordering.

of the chiral rollers' trajectories increases (with the strength of the applied electric field) the persistence length (and as a result the interaction length scale) decreases, and the circular motion of rollers becomes more localized. The rollers cannot sustain large-scale vortical patterns anymore, and transition to a rotating flocks phase still preserving local velocity and phase synchronization. Further decrease of the interaction length scale of the rollers, driven by the increase of the curvature of their trajectories with the field strength, leads to the loss of the velocity alignment and phase synchronization, and the system transitions to a gas of spinners.

The system of chiral Quincke rollers is similar to the generic Quincke roller systems with spherical particles except the shape

anisotropy. As was discussed previously[27], the alignment mechanisms come from both hydrodynamic and electrostatic interactions with the dominance of hydrodynamic interactions. While electrostatic alignment interactions scale as $\sim(1 - E_q^2/E^2)$ (here $E_q$ is a constant dependent on the particle and media properties for spherical Quincke rollers) the hydrodynamic interactions yield no dependence on the strength of the electric field[27]. For the shape-anisotropic Quincke rollers the effect of the electric filed on alinement interactions is more complicated since the strength of the field also controls the curvature of rollers' trajectories, and thus the persistence length and the localization of the rollers. As a result, the effect of the electric field on the collective behavior is non-monotonic that is evident from the

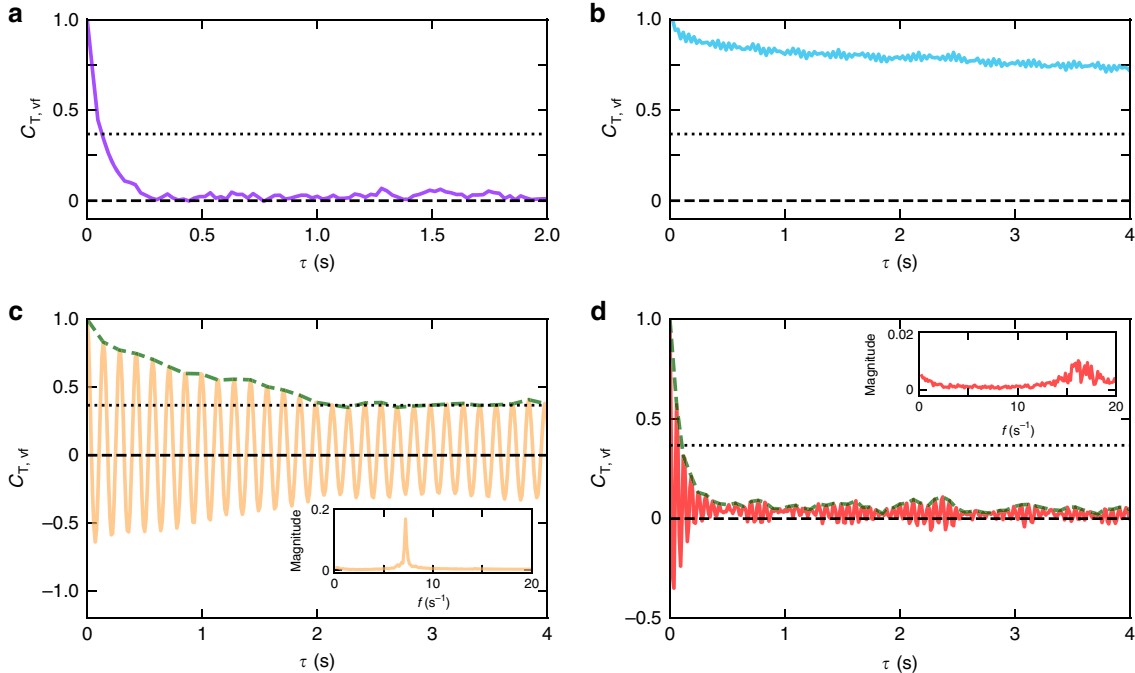

**Fig. 5 Temporal correlations of the velocity fields for different dynamic phases. a** Gas. $E = 1.60\ \mathrm{V\ \mu m^{-1}}$. $\phi = 0.054$. **b** Vortices. $E = 1.92\ \mathrm{V\ \mu m^{-1}}$. $\phi = 0.108$. **c** Rotating flocks. $E = 2.60\ \mathrm{V\ \mu m^{-1}}$. $\phi = 0.144$. **d** Spinners. $E = 2.80\ \mathrm{V\ \mu m^{-1}}$. $\phi = 0.054$. Inserts demonstrate FFTs of corresponding $C_{\mathrm{T,vf}}$ curves in rotating flocks and spinners phases. Dotted lines indicate $1/e$ value. Green dashed lines in (**c, d**) show the positive envelopes of the correlation functions.

presence of uncorrelated phases at both lower (Gas) and higher (gas of Spinners) field strengths, and collective phases (Vortices and Rotating flocks) observed in the intermediate region of the field strengths, see Fig. 3a.

**Characterization of emergent patterns.** To investigate and quantify changes in the behavior of chiral rollers with the activity, we employ velocity temporal correlation function:

$$C_{\mathrm{T}}(\tau) = N^{-1} \sum_i \langle \mathbf{v}_i(t) \cdot \mathbf{v}_i(t+\tau) \rangle_t / \langle \mathbf{v}_i^2(t) \rangle_t. \qquad (1)$$

Here, $\mathbf{v}_i(t)$ is the velocity of a chiral roller $i$ at the moment $t$; $N$ is the total number of rollers; and $\langle\ \rangle_t$ indicates time averaging. In Fig. 3b, we show corresponding correlation curves obtained for different activity levels. Analogous to $\Delta r^2(\tau)$, $C_{\mathrm{T}}$ develops strong and long-lasting oscillations with the activity due to continuous rotation of rollers on circular trajectories of higher curvatures (Fig. 3b). Rollers preserve a narrow distribution of the frequency of oscillations that increases with the activity (Fig. 3b insert). The functional dependence of the oscillation frequency is nontrivial as it depends on both, increase of a roller's velocity with the electric field strength, and change of the curvature of the particle trajectory defined by the particle geometry and shape asymmetry.

The emergence of spatial coherence in our system is tracked by the analysis of the velocity spatial correlation function:

$$C_s(r) = \frac{\left\langle \left\langle \mathbf{v}_i(r_0, t) \cdot \mathbf{v}_j(r_0 + r, t) \right\rangle_{i,j} \right\rangle_t}{\left\langle \left\langle \mathbf{v}_i^2(r_0, t) \right\rangle_i \right\rangle_t}, \qquad (2)$$

where $\mathbf{v}_i(r_0, t)$ and $\mathbf{v}_j(r_0 + r, t)$ are velocities of roller $i$ and $j$ with a relative distance $r$ at a time $t$, $\langle\ \rangle_i$ is an ensemble average, and $\langle\ \rangle_t$ describes averaging over experimental realizations. At low $E$, corresponding to an uncorrelated gas of chiral rollers, $C_s$ rapidly decays to 0 (Fig. 3c). As $E$ increases, the spatial correlations between chiral rollers grow, and vortices emerge in the system triggering characteristic anti-correlation regions in the

corresponding $C_s$ curves. Further increase in $E$ leads to a gradual disappearance of the anticorrelations as system transitions to a rotating flocks phase. A typical evolution of the spatial correlation length, $L_c$, with the electric field is shown in the inset of Fig. 3c. $L_c$ is defined as the distance where $C_s$ decays to $1/e$ (see Supplementary Fig. 3). The maximum coherence between chiral rollers is observed at the vortex phase.

A characteristic vortex size, $D_v$, defined as a second zero crossing in the spatial correlation curve $C_s(r)$ (see Supplementary Fig. 3), has a similar non-monotonic trend (Fig. 3e). Consequently, rollers with longer persistence length $L_p$ form larger vortices and the overall $D_v$ dependence on the persistence length, $L_p$, is roughly linear (Fig. 3f). Both properties reach their maximums at a verge of transition ($\kappa \sim 0$, Fig. 1d) between $\alpha$ and $\gamma$ regimes of the chiral rolling indicating a strong link between a chiral behavior of rollers, activity and emergent self-organized patterns.

**Local order inside of collective states.** Remarkably, the emergence of a spontaneous swirling motion in a system of chiral pear-shaped Quincke rollers is also accompanied by a distinctive structural ordering inside the vortices and flocks controlled by the activity. Three distinctive types of vortices are revealed in the system. $\alpha$-vortices are formed with all the rollers aligning their directions away from the vortex 'eye' ('heads-out' order). In $\gamma$-vortices rollers' orientations coherently point toward the center of the vortex ('heads-in' order). $\beta$-vortices are comprised of both 'heads-in' and 'heads-out' rollers. Figures 3g–h and 4 illustrate the internal alignments of the chiral rollers inside the collective phases.

$\alpha$-vortices are realized for $\kappa < 0$, when $E$ is below the curvature transition region of the chiral rollers, while $\gamma$-vortices populate regions with $\kappa > 0$ (see Fig. 1d). In both scenarios anisotropic pear-shaped rollers align their particle's orientations and velocity directions with the neighboring rollers to move collectively in a swirling pattern. At the transition region corresponding also to a

longest persistence length, $L_p$, the curvature parameters, $\kappa$, transitions through zero indicative that both modes of the chiral roller motion ('heads-in' and 'heads-out') are equally probable to be found in the system due to unavoidable slight polydispersity (shape, size, and surface chemistry) of the chiral rollers, and as a result emergent collective vortices are comprised of rollers of both orientations as illustrated in Fig. 4c. In the case of $\alpha$- and $\gamma$-vortices, the system spontaneously separate rollers rotating CW and CCW to different vortices with opposite sense of rotation. A similar process has been observed in simulations of chiral particles on a surface[33].

To quantify the level of ordering in the collective phases of chiral rollers, we calculate two locally defined order parameters: an orientational polar order parameter, $\Psi_1$, and a nematic order parameter, $\Psi_2$:

$$\Psi_{n,i} = \frac{1}{N_i} \sum_{j=1}^{N_i} \exp(in\varphi_j). \tag{3}$$

Here, $\varphi_j$ is the in-plane projection of a roller's orientation, $N_i$ is the number of rollers in a local square grid cell $i$, see 'Methods' for the details. $n = 1$ corresponds to a polar order parameter, while $n = 2$ describes a nematic order. In Fig. 4f, we show a probability distribution functions (PDF) of $|\Psi_1|$ and $|\Psi_2|$ obtained for different collective phases of the chiral rollers ensemble. For $\alpha$- and $\gamma$-vortices, both PDF($|\Psi_1|$) and PDF($|\Psi_2|$) reveal strong peaks close to $|\Psi_{1,2}| = 1$, indicative of a polar orientational order of the chiral rollers. In contrast, only the PDF($|\Psi_2|$) obtained for the $\beta$-vortices exhibit a main peak close to the $|\Psi_2| = 1$, while the corresponding peak of the PDF($|\Psi_1|$) at $|\Psi_1| = 1$ is suppressed, indicative of mostly nematic local ordering inside of the $\beta$-vortices. Several satellite peaks in the middle of the above distributions originate from the local disorder (see Supplementary Note 2 and Supplementary Fig. 4). As the system transitions into a the rotating flocks state, strong polar order emerges as demonstrated by the PDF for $|\Psi_1|$ shown in Fig. 4f.

## Discussion

In conclusion, we demonstrate that an active system comprised of shape-anisotropic rollers gives rise to a active chiral fluid capable of complex collective behavior and self-organization often not available in the case of spherical rollers. Pear-shaped Quincke rollers exhibit activity dependent particle orientations and curvature of their trajectories without externally prescribed handedness of their motion. By controlling the activity in the system through variations of the energizing electric field, we reveal a set of emergent reconfigurable structures: gas of spinners, localized aster-like vortices and rotating polar flocks, enabled by the chiral motion of the unitary building blocks (circle rollers). Among the remarkable features of these structures is the ability to develop and manipulate on demand the particles' collective orientational order, polar, or nematic, in response to the activity levels. Arrays of self-organized vortices show tendency toward a 2D antiferromagnetic ordering. Our work provides insights into the onset of spatial and temporal coherence in active chiral systems, and hints at design pathways for active self-assembled architectures with externally controlled reconfigurability.

## Methods

**Experimental details**. Pear-shaped polystyrene particles (PNT010UM, Magsphere Inc.; long axes $d_l = 10.5\,\mu m$, short axes $d_s = 9.0\,\mu m$) are dispersed in a 0.15 mol $L^{-1}$ AOT/hexadecane solution. The average particle size $d = (d_l + d_s)/2 = 9.8\,\mu m$. The colloidal suspension is injected into a cylindrical chamber composed of a SU-8 toroid and two parallel ITO-coated glass slides (IT100, Nanocs). The thickness and inner diameter of the SU-8 toroid are 40 $\mu m$ and 2 mm, respectively. The electric field is supplied by a function generator (Agilent 33210A, Agilent Technologies) and a power amplifier (BOP 1000M, Kepco Inc.). The field strength $E$ is varied

between 1.4 and 2.8 V $\mu m^{-1}$. The Reynolds number, $Re \simeq 0.005$ for a roller with a typical speed $|\mathbf{v}| = 1\,mm\,s^{-1}$.

The sample cell is observed under a microscope with a 4× microscope objective. Videos are recorded by a fast-speed camera (IL 5, Fastec Imaging) at 430 frames per second (FPS). In a typical experiment, once the electric field is on, rollers self-organize into the corresponding patterns and reach a steady state within a few seconds. We capture 8-s videos to analyze the structure and dynamics of the particles 1 min after the field has been switched on. During a typical run time of the experiments (about 10 min), no apparent coarsening of the patterns has been observed. Particle tracking velocimetry (PTV) and further data analysis are carried out with custom codes in Python and Trackpy[52].

**Discrimination of different phases**. To quantitatively discriminate between gas of spinners and rotating flocks phases in the phase diagram, the temporal correlations function of the velocity field is used:

$$C_{T,vf}(\tau) = N^{-1} \sum_i \langle \mathbf{v}_i(0) \cdot \mathbf{v}_i(\tau) \rangle_t / \langle \mathbf{v}_i^2(0) \rangle_t. \tag{4}$$

The system is divided into a set of square grid cells with sizes of $4d$ to calculate the velocity field. $\mathbf{v}_i$ is the average velocity at a cell $i$; $N$ is the total number of cells; and $\langle\,\rangle_t$ indicates time averaging. Note that $C_{T,vf}$ quantifies the temporal correlation of the velocity field at defined at fixed positions while $C_T$ deals with the temporal velocity correlations of a roller. Figure 5 demonstrates typical $C_{T,vf}$ curves calculated for different phases. In the gas state (Gas and Spinners phases), $C_{T,vf}$ falls sharply to 0 while $C_{T,vf}$ curves for collective phases (Vortices and Rotating flocks) maintain much slower time decay. Quantitatively, we use the following criterion to discriminate between rotating flocks and spinners phases that both exhibit strong oscillatory behavior of $C_{T,vf}$ (see Fig. 5c, d): if within ten oscillations of the temporal correlation of the velocity field, $C_{T,vf}$, the positive envelope of the correlation function is above $1/e$ ($e$ is Euler's number) the phase is considered as collective (rotating flocks) otherwise it is a spinners phase.

**Calculation of the local orientational order parameters**. For the calculations of the local orientational order parameters $|\Psi_1|$ and $|\Psi_2|$, the system is divided into a set of square grid cells with sizes of $6d$ (see Supplementary Note 2 and Supplementary Fig. 4). The procedure is analogous to one described in ref. [53].

## Data availability
The data in support of the reported findings are available from the corresponding author upon request.

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

## Acknowledgements

The research was supported by the US Department of Energy, Office of Science, Basic Energy Sciences, Materials Sciences and Engineering Division. Use of the Center for Nanoscale Materials, an Office of Science user facility, was supported by the US Department of Energy, Office of Science, Office of Basic Energy Sciences, under Contract No. DE-AC02-06CH11357.

## Author contributions

A. Snezhko and B.Z. conceived the research. B.Z. performed the experiments. B.Z., A. Sokolov and A. Snezhko analyzed the data and wrote the paper.

## Competing interests

The authors declare no competing interests.
