## [Peer Review File · Nature Communications]

Reviewers' comments:

Reviewer #1 (Remarks to the Author):

The authors experimentally realize and explore a system of asymmetric (pear-shaped) Quincke rollers which move in circles along a substrate. Unlike spherical Quincke rollers, the asymmetric rollers do not rotate around a random axis perpendicular to the applied electric field, but predominantly rotate around their long-axis. Following their shape-asymmetry also w.r.t. to this long-axis the particles experience a viscous extra-torque leading to circular trajectories. Here chirality is broken spontaneously on the level of each roller and correspondingly, clockwise and counterclockwise rotations are simultaneously present in the experiment.

The authors systematically study the phase diagram of the circle-rollers and find a rich collective behavior, which they characterize in detail mostly based on velocity correlation functions: At low activity the system is in the gas-phase. If the density is not too low, at higher activity (larger E) where the particles move faster but feature a decreasing rotation radius (due to a nontrivial change of the tilt-angle), the system first forms vortices and for higher E also rotating flocks. At low density and high activity, the particles rotate in place and form a "spinner" phase.

The present work provides a nice new experimental realization of chiral active matter, providing direct experimental evidence for previously predicted phenomena such as rotating flocks and spontaneous phase separation of CW and CCW rotating particles. Accordingly, this work should be of immediate interest to researchers working e.g. on chiral active matter, flocking, or more generally on pattern formation in active systems. The experiment also promises interesting generalizations in the future, e.g. of mixtures of circle rollers.

Accordingly, from my viewpoint the present manuscript is generally suitable for Nature Communications. However, before finally recommending its publication I would like to ask the authors to consider the following points:

(i) How are the individual phases in Fig. 3a precisely defined? What is the precise criterion discriminating e.g. between the spinner phase and the gas phase?

(ii) Can the author say something about the mechanism leading to the transition from aster-like states to rotating flocks in their experiment? If not, they might want to mention this as a relevant open point in their conclusions.

(iii) What is the precise origin of the alignment interactions in the present setup? Do they have a purely hydrodynamic origin or is there also an electrostatic alignment contribution? How does the alignment strength scale with E ?

(iv) Regarding the timescale of the experiment.

Both the movies and Fig. 3b suggest that the experiments run for about $\sim 10^2$ circulation periods of the particles.

Could the authors provide an explicit statement in their manuscript regarding the run time of the experiment?

Related to this: The authors indicate that they do not see any coarsening of the rotating flocks in the course of their experiment. I guess that this is due to the relatively short time the experiment is running. Or would the authors expect to see no coarsening also at much larger timescales?

(v) Note that the observed spontaneous segregation of CW and CCW rotating chiral active particles has been previously observed in simulations of minimal models of chiral active particles, see e.g. Phys.

Rev. E 100, 012406 (2019).

(vi) The phase diagram in Fig. 3 is shown up to $E \sim 3 V/\mu m$. What would happen for $E > 3V/\mu m$? Is there an experimental limitation making it impossible to explore this regime?

(vii) It could be interesting to quantify the polarization within the rotating flocks. How does the polarization change (qualitatively or quantitatively) when enhancing E ?

(viii) The asymmetric Quincke-rollers might deserve a name such as e.g. "circle-rollers" akin to colloidal "circle swimmers".

(ix) The caption of Fig. 3 is too long I think. I suggest showing legends within some of the panels of Fig. 3 to make the figure better accessible.

Reviewer #2 (Remarks to the Author):

Zhang et al report the emergence of two classes of dynamical patterns in collections of self-propelled colloidal rollers. Actuating polar colloidal dumbbells using an electrohydrodynamic instability, the authors show how to control both the translational and orbital speed of model self-propelled bodies. They provide a characterisation of their individual dynamics before addressing their collective behavior. When the polar rollers interact they can either self assemble into collection of vortices, or into a nonequilibrium phase termed rotating flocks. In rotating flocks, all rollers orbit in synchrony over macroscopic regions of space propelling along circular trajectories. The transition between the two dynamical regimes is controlled by the magnitude of the electric field responsible for the so-called Quincke rotation at the origin of self-propulsion.

The reported phenomena are truly spectacular and very carefully characterized by a number of quantitative measurements (which is unusual in the field). I have however two concerns about this work:

— Firstly, the dynamics of orbiting active particles is not new. They were reported in earlier experiments from the Oiwa and Granick groups and theoretically/numerically discussed by the Chaté and Luijten groups (among others). These reference should be cited in the main text and the presented results thoroughly compared to these earlier reports.

Effective temperature concept evaluated in an active colloid mixture

By Ming Han, Jing Yan, Steve Granick, and Erik Luijten PNAS July 18, 2017 114 (29) 7513-7518

Large-scale vortex lattice emerging from collectively moving microtubules, Nature volume 483, pages 448–452 (2012)

By Yutaka Sumino, Ken H. Nagai, Yuji Shitaka, Dan Tanaka, Kenichi Yoshikawa, Hugues Chaté & Kazuhiro Oiwa

In fairness, Zhang et al provide cleaner and more extensive experimental data, and their system makes it possible to continuously explore the phase behavior of the interacting colloids (This was also possible with in the experiments by Ha and coworkers, but not discussed as thoroughly as in the manuscript of Zhang et al).

— Secondly, while the experimental characterisation of the emergent dynamics is very neat, the authors fall short of explaining the rich phenomenology they observed. As a consequence, the

manuscript is plagued with rather vague statements about the origin of the various dynamics (one-body and many-body dynamics).

Reading this manuscript, I could not understand what are the mechanisms underlying the dynamical transitions observed between the Gas, Vortex and Rotating flock regimes upon increasing E . How are the symmetries and magnitude of the interactions between the particles altered by E ? Is the single-trajectory curvature enough to explain the full phase behavior? Are the three regimes, three genuinely distinct dynamical phases? Are the domains of the phase diagram separated by smooth crossovers? Can the observed phenomenology be accounted for by the Vicsek-like models studied e.g. in Activity induced synchronization: Mutual flocking and chiral self-sorting, Phys. Rev. Research. 2019 by Demian Levis, Ignacio Pagonabarraga, and Benno Liebchen?

Given the absence of clear comparisons with the abundant earlier literature (experiments, numerics and theory), and the lack of explanations for the rich collective dynamics observed in these beautiful experiments, I am afraid I cannot recommend the current manuscript for publication in Nature Communications.

More specific questions and comments:

— What does set the location of the vortices? Does it change from one experiment to another (in the same device), or do they reflect some built-in heterogeneities? What is responsible for the cohesion of these vertical structures?

— When $\kappa=0$, I would have expected a standard flocking phase to emerge (instead of β vortices). Could the authors explain why it is absent from the phase diagram?

Does this reflect the polydispersity of the curvature distribution that does not peak at 0 when the $\langle \kappa \rangle = 0$?

— At the single-colloid level, could the author provide a qualitative explanation for the chirality reversal? (a quantitative explanation would be even better)

— Fig. 1b. Do the authors plot the most probable or the average speed value. The suppression of a sharp bifurcation could well be a mere artifact caused by some polydispersity in the particle shape/chemistry. Plotting the most probable value of the propulsion speed should suppress this possible artifact.

— Terming κ "curvature" might be misleading, $1/\kappa$ is not a length scale.

— As Fig. 1d hardly depends on E , I feel that a 2D plot of $\kappa(E)$ could provide a clearer illustration of the change in the roller behavior.

Reviewer #3 (Remarks to the Author):

This experimental work investigates the patterns of spontaneous rotations performed by colloidal pear-shaped dielectric particles confined in a cylindrical cell and powered by a constant electric field. The experimental results show very attractive and to my knowledge novel collective behavior; the study covers

an extensive range of quantities and parameters. However, the manuscript is written in an unclear manner, and after reading it quite carefully, I think there are contradictions, and several superficial or even might be wrong statements. Unfortunately, I can therefore not recommend this work for publication, at least not in its actual stage.

Following I include some more detailed comments:

1.- First, the abstract and introduction do not give clear hints of what is done in the manuscript and have in fact very little information about. In the conclusion paragraph, it is only the first few lines that summarize the work, while the second half refers to unproven and I would say unlikely projections of their results.

2.- In the first part of the manuscript the dynamics of individual particles is analyzed (around Fig. 1). The motion of the colloid is described as belonging to three modes attending to the orientation of the main axis with respect to the substrate plane. Figure 1c, and the sentence: "Transitions from α to γ modes with the activity also inevitably lead to a chiral states reversal of the individual rollers" indicate that the trajectory of mode β is straight, while the other two are curved, so one would be CW and the other CCW but which one is which one? And we can understand that fixing the applied field and the density these could not coexist then. Is this correct?

3.- I find the discussion of the MSD and MSAD shown in Figs 1e and 1f quite confusing. 3a.-- The text only states: "Initially the rollers move ballistically as $\text{msd} \sim \tau^2$ and then transition to a diffusive-like regime at longer times."

As far as I see this is clear the case only for the trajectory of mode β , since mode α seems to remain ballistic much longer and mode γ can become even subdiffusive.

3b.-- The text further states: "At high activity levels, the curvature of the trajectories increases resulting in an appearance of characteristic oscillations in the mean square displacement". But the oscillations are only appearing in the mode β data, so should we understand that this mode is the only one showing rotating trajectories?

3c.-- The MSAD data seems to agree with the previous statement since mode γ remains ballistic, while the other data clearly show the transition from diffusive to ballistic, so this would indicate rotational motion to all modes, contradicting previous statements

4.- Later in the manuscript collective dynamics is investigated and linked to the single particle properties. Although this would be a standard solid procedure to understand the system, here is confusing. In page 4, and related to date in Fig. 2e it is stated that: "the system of pear-shaped rollers exhibits a spontaneous chirality induced phase separation from initially random distribution of chiral rollers"

The initially random distribution is absolutely not proved here, and also not the transition but only an already phase separated system. Looking very carefully at all the provided movies, I would say that it seems more that the rotation is more dictated by the environment, (Visek type of behavior) and not to the individual properties as discussed here.

5.- Discussing Fig. 2f the authors refer to the subdiffusive behavior shown in Fig. 1e, but as far as I understand these two sets of data correspond to different densities, so no linked or conclusion could be made. The MSD at higher densities should then be also

measured, and the collective behavior at more dilute regimes discussed.

6.- The radius of the individual trajectories seem to be importantly affected by the intensity of the applied field and/or by the density, as shown in Figs 2c and 2f.

I find this very interesting, but it has not been even discussed at the single particle regime level, and I think should be investigated for all range of E and ϕ values here presented.

7.- Is the velocity in Eq.(1) related with the ballistic velocity of Fig 1e, and the frequencies in the inset of Fig 3b are probably related with the angular velocities of Fig 1f ?

If there is a reason not to link these values it should be explained, otherwise a more in deep analysis should be presented. For example the frequencies are claimed to increase linearly with activity, but how linearly, quadratically, do the authors present an argument of why is this behavior expected ?

8.- The persistence length L_p is here defined "as the distance particles travel where the velocity temporal correlation function C_t decays to $1/e$ ". I assume this has to be equivalent, or at least related to the more intuitive and more commonly used definition of the decay of the orientation correlation along the trajectory as most, is this the case ?

All movies and the trajectories shown in Figs. 3c, 3f show that the distribution of such L_p 's should be broad, since it can be very different depending on the particle relative position to the vortex center or boundary area. This is not discussed and I think it will have a big influence in the relevance of these quantities.

9.- Are all the points in Fig 3a represented in Fig 3f ? To keep the symbols then could bring some additional information.

10.- I find very interesting that the nature of the vortices changes with the intensity of the applied field, but I have been quite confused about it.

The reader is confronted to vortices α , γ that rotate always CCW, while the only movie of various coexisting vortices shows both CCW and CW rotations.

In the end, I realized that Fig 4c corresponds to Fig 3a, and Fig. 4e to Fig 3d, while no general image is shown for Figs 4a, 4b, and 4d. To discuss this and show the corresponding movies, and plots of the velocity-vorticity and trajectories seems necessary to me.

11.- I also assume now that the beta vortex is able to choose both CW and CCW, and therefore is this type (with its very limited set of possible parameters) the only one able to show the discussed coexisting vortices of opposite rotations. In the text now, it seems that this behavior is more the rule than the exception.

12.- On the other hand, I am not sure if this contradicts the discussion at the beginning of the manuscript were the rotation direction was stated to be determined the applied field (see Fig. 1d). The rotations we supposed to have opposite direction for α and γ , while non-existing for β . It could be that the explanation for this is that the effect emerges with increasing density, but something should be said and probably characterized about it, or simply that the single particle motion need more clarification.

13.- The authors also claim around Fig. 2 that the discussed patterns are quite stable on time. However, there is not much information about for how much time is this checked, or if this happens for all types of patterns, or only for the two shown in Fig. 2.

14.- The spinner phase is only shown in the most diluted case. I would find interesting to

see how this is for higher densities.

15.- If I understand properly the PDF's of ψ_1 and ψ_2 shown in the manuscript and in more detail in the supplementary note correspond just to a very particular configuration, this is one realization at one particular time.

This would mean that these peaks can change with time and would disappear if an averaged PDF would be shown. Would the rest of the distribution then remain stable in that case, and I would wonder about the relevance of the peaks in a general discussion of the system as the one presented in this work. More important would be to discuss the polar order shows two maximums in the vortex phase, but not the nematic order.

16.- Finally I am interested on the importance of the system size.

Would it possible to make experiments with cell of different sizes ?

Both smaller or larger would provide important information. In case this is not possible, a discussion of the possible outcome, and the consequent universality of the obtained phase diagrams seems relevant here.

Other minor comments the authors might be interested on:

- Fig 1e refers to solid line as having slope 1, but all data are also solid lines.
- The frequently referred purple color is a bit darker blue in my prints.
- Caption Fig 3a: I guess that the colors do not refer to the symbols as stated in the caption, but to the background color.
- The scale bars would be more clear referring to the same length in all plots.
- Why there is no data for $\phi=0.001$ in Fig, 3e ?
- Symbols are not always defined before being used.
 - For example 'd' only appears in the method section, but is used in Fig.3
 - E and E0 are first used and a bit later only E is introduced.

Reply to Reviewer 1

We thank the Reviewer for a careful reading of our manuscript and positive judgement. The Reviewer made a number of comments and suggestions aimed to improve the clarity of our paper, which we fully addressed in the revised manuscript. Below are point-by-point replies to the Reviewer's comments:

(i) How are the individual phases in Fig. 3a precisely defined? What is the precise criterion discriminating e.g. between the spinner phase and the gas phase?

Indeed, while some of the phase can be distinguished easily by the dynamics of the emergent patterns (such as vortices versus rotating flocks) or structural orientational order inside the patterns (α , β or γ vortices), the other phases, such as those mentioned by the Reviewer, require a discriminating criterion to separate. At low densities rollers are in uncorrelated gas state. We further discriminate rollers in the gas phase with the γ mode of motion (above 1.96 V/um) and call them spinners as they turn into localized spinners at high strength of the electric field.

The transition region between rotating flocks and spinners (gas of spinners) is not sharp as flocks gradually fall apart as we decrease rollers density at a fixed activity (controlled by the electric field strength). To quantitatively discriminate between these two phases in the phase diagram, we involve the temporal correlation functions of the velocity field, $C_{t,vf}(\tau) = N^{-1} \sum_i^N \langle v_i(0) \cdot v_i(\tau) \rangle_t / \langle v_i^2(0) \rangle_t$. The system is divided into a set of square grid cells with a size of $4d$ (d is a particle diameter), and average v_i is calculated for each cell i ; N is the total number of cells; and $\langle \rangle_t$ indicates time averaging. Note that $C_{t,vf}$ quantifies the temporal correlation of the velocity field at fixed positions while C_t (also used in our manuscript) deals with the temporal velocity correlation of a roller. New Fig. 5 shows typical $C_{t,vf}$ curves for different phases. In the gas state (Gas and Spinners phases), $C_{t,vf}$ falls sharply to 0 while $C_{t,vf}$ curves for collective phases (Vortices and Rotating flocks) maintain much slower time decay. Quantitatively, we use the following additional criterion to discriminate between rotating flocks and spinners phases that both exhibit strong oscillatory behavior of $C_{t,vf}$ (as shown in new Fig 5 panels c,d): if within 10 oscillations of the temporal correlation of the velocity field, $C_{t,vf}$, the positive envelope of the correlation function is above $1/e$ (e is Euler's number) the phase is considered as collective (rotating flocks) otherwise it is a spinners phase.

We added corresponding discussion and new Fig 5 in the revised main text and the Methods section.

(ii) Can the author say something about the mechanism leading to the transition from aster-like states to rotating flocks in their experiment? If not, they might want to mention this as a relevant open point in their conclusions.

Formation of collective phases in our active system is driven by velocity alignment interactions. The alignment mechanism due to inelastic particle collisions is discussed in literature in detail in several publications which we reference in the revised manuscript Refs. 49-51. In a few words, inelastic collisions between active particles lead to velocity alignment and build-up of spatial correlations. In particular, for the systems of Quincke rollers both hydrodynamic and electrostatic interactions contribute to the velocity alignment as was discussed in detail in Ref. 27. In the case of chiral rollers the vortical (aster-like) patterns emerge at the transition region between two mods of rolling and correspond to largest persistence lengths for individual rollers (low curvature of trajectories), that facilitate the large interaction range (of the order of the system size) and time for rollers to interact and enhance their alignment. The sizes of vortices are controlled by the interaction range as demonstrated by the linear relation between the vortex sizes and the persistence length shown in Fig. 3f. As the curvature of the chiral rollers' trajectories increases (with the strength of the electric field) the persistence length (and as a result the interaction length scale) decreases, and the circular motion of rollers becomes more localized.

The rollers cannot sustain a large-scale vortical pattern anymore and transition to a rotating flocks phase still preserving local velocity and phase synchronization. Further decrease of the interaction length-scale of the rollers (due to increase of the curvature of their trajectories with the field strength) leads to the loss of the velocity alignment and phase synchronization- the system transitions to a gas of spinners.

We added corresponding discussion in the revised text. We also added in the conclusion that further rigorous theoretical/computational studies are needed to develop detailed understanding of the processes leading to pattern formation and synchronization in the systems with activity dependent chiral motion of particles.

(iii) What is the precise origin of the alignment interactions in the present setup? Do they have a purely hydrodynamic origin or is there also an electrostatic alignment contribution? How does the alignment strength scale with E ?

Our system is similar to the generic Quincke roller system with spherical particles except the shape-anisotropy. As was discussed in detail in Ref. 27, the alignment mechanisms come from both hydrodynamic and electrostatic interactions (that scales as $\sim (1-E_c^2/E^2)$ for spherical Quincke rollers) with the dominance of hydrodynamic interactions that yield no dependence on the strength of the electric field.

Due to the shape-anisotropy in the system of chiral Quincke rollers, the effect of the electric field on alignment interactions is more complicated and pronounced since the strength of the field now controls the curvature of a roller's trajectory, and thus the persistence length and the interaction range of the roller. As a result, the alignment strength has a non-monotonic dependence on the electric field strength: uncorrelated phases at both lower (Gas) and higher (gas of Spinners) field strengths and collective phases (Vortices and Rotating flocks) in the intermediate region of the field strengths.

(iv) Regarding the timescale of the experiment.

Both the movies and Fig. 3b suggest that the experiments run for about $\sim 10^2$ circulation periods of the particles. Could the authors provide an explicit statement in their manuscript regarding the run time of the experiment?

Related to this: The authors indicate that they do not see any coarsening of the rotating flocks in the course of their experiment. I guess that this is due to the relatively short time the experiment is running. Or would the authors expect to see no coarsening also at much larger timescales?

In a typical experiment, once the electric field is on, rollers self-assemble into the corresponding patterns and reaches a steady state within a few seconds. We start to capture 8-second videos to analyze the structure and dynamics of the particles 1 min after the electric field has been switched on. During a typical run time of the experiments (about 10 min), no apparent coarsening of the rotating flocks has been observed. It is possible that the coarsening dynamics has much larger timescale (hours and days). We added corresponding information in the revised text.

(v) Note that the observed spontaneous segregation of CW and CCW rotating chiral active particles has been previously observed in simulations of minimal models of chiral active particles, see e.g. Phys. Rev. E 100, 012406 (2019).

We thank the reviewer for bringing to our attention the recent relevant reference. We are aware of works from D Levis and B Liebchen and cited previously their earlier related work on the topic (PRL 119.5 (2017)). We included the reference suggested by the referee in the revised text.

(vi) The phase diagram in Fig. 3 is shown up to $E \sim 3 \text{ V}/\mu\text{m}$. What would happen for $E > 3 \text{ V}/\mu\text{m}$? Is there an experimental limitation making it impossible to explore this regime?

Some of the experiments were performed in fields up to $E = 4 \text{ V}/\mu\text{m}$. No new phases/dynamics have been observed above $E \sim 3 \text{ V}/\mu\text{m}$. The rotating flocks phase turns into spinners phase with the increase of the electric field strength. We selected the current cutoffs in the phase diagram as it captures all major phases observed in this chiral roller ensemble.

(vii) *It could be interesting to quantify the polarization within the rotating flocks. How does the polarization change (qualitatively or quantitatively) when enhancing E ?*

Indeed, we agree with the reviewer that it would be interesting to further investigate in detail the dynamics and degree of polarization within the rotating flocks. With the increase of the field strength within the boundaries of the rotating flocks phase the polarization of the flocks first increases (when we move away from the phase boundary with the vortex phase) and then decreases due to shrinking interaction radius of the chiral rollers (high curvature of the trajectories) and higher localization when approaching a phase boundary with the spinner phase. We added a Supplementary Figure S2 (accompanied by a Supplementary Note 1) showing scalar nematic order parameter and corresponding probability distribution functions of the particle orientations within the flocks at different strengths of the electric field. We added corresponding comment into the revised text.

(viii) *The asymmetric Quincke-rollers might deserve a name such as e.g. "circle-rollers" akin to colloidal "circle swimmers".*

We added corresponding note in the revised text.

(ix) *The caption of Fig. 3 is too long I think. I suggest showing legends within some of the panels of Fig. 3 to make the figure better accessible.*

Following Referee's recommendations, we added additional legends within some of the panels of Fig.3.

Reply to Reviewer 2

We thank the Reviewer for a careful reading of our manuscript and finding our results novel, "truly spectacular", and "carefully characterized". The Reviewer made a number of comments and suggestions aimed to improve the introductory and explanatory part of our paper which we fully addressed in the revised manuscript. Below are point-by-point replies to the Reviewer's comments:

1. Firstly, the dynamics of orbiting active particles is not new. They were reported in earlier experiments from the Oiwa and Granick groups and theoretically/numerically discussed by the Chaté and Luijten groups (among others). These reference should be cited in the main text and the presented results thoroughly compared to these earlier reports.

Effective temperature concept evaluated in an active colloid mixture. By Ming Han, Jing Yan, Steve Granick, and Erik Luijten PNAS July 18, 2017 114 (29) 7513-7518

Large-scale vortex lattice emerging from collectively moving microtubules, Nature volume 483, pages 448–452 (2012). By Yutaka Sumino, Ken H. Nagai, Yuji Shitaka, Dan Tanaka, Kenichi Yoshikawa, Hugues Chaté & Kazuhiro Oiwa

We agree with the reviewer that the above references are highly relevant. In the revised version of the manuscript we cite those references, and make comparison to our system as suggested by the Reviewer. In addition, we modified the introductory part of the manuscript to provide more substantial overview of the current state of the field on chiral active particles.

2. Secondly, while the experimental characterisation of the emergent dynamics is very neat, the authors fall short of explaining the rich phenomenology they observed. As a consequence, the manuscript is plagued with rather vague statements about the origin of the various dynamics (one-body and many-body dynamics).

Reading this manuscript, I could not understand what are the mechanisms underlying the dynamical transitions observed between the Gas, Vortex and Rotating flock regimes upon increasing E . How are the symmetries and magnitude of the interactions between the particles altered by E ? Is the single-trajectory curvature enough to explain the full phase behavior? Are the three regimes, three genuinely distinct dynamical phases? are the domains of the phase diagram separated by smooth crossovers?

Indeed, we agree with the Referee that the manuscript will benefit from more discussions on the mechanisms underlying the observed rich phenomenology in our system. In the revised version of the manuscript we introduced new paragraphs discussing underlying alignment interactions and mechanisms driving transitions between different phases. For the systems of Quincke rollers both hydrodynamic and electrostatic interactions contribute to the velocity alignment as was discussed in detail in Ref.27. Please see the reply to the second question of Referee 1.

For the discussion on how interaction between particles scale with the electric field strength please see also the reply to the question (iii) from the Reviewer 1. We added corresponding comments in the revised text.

The curvature of a particle trajectory and the particles area density is enough to reproduce the full phase behavior. While all phases are dynamically very distinctive, the crossovers between them are smooth involving transition regions (both phases are simultaneously present). The criteria used to define different phase boundaries are now describe in the main text and the Methods section.

Can the observed phenomenology be accounted for by the Vicsek like models studied e.g. in Activity induced synchronization: Mutual flocking and chiral self-sorting, Phys. Rev. Research. 2019 by Demian Levis, Ignacio Pagonabarraga, and Benno Liebchen?

We thank the Reviewer for bringing our attention to this reference. Some of the observed phenomenology like rotating flocks and spinners, chirality induced phase separation can be accounted for by the Vicsek-like models (the work suggested by the Reviewer and two previous works from Benno Liebchen and Demian Lewis that we cited previously). We now cite all three works on the matter.

3. *What does set the location of the vortices? Does it change from one experiment to another (in the same device), or do they reflect some built in heterogeneities? What is responsible for the cohesion of these vertical structures?*

The locations of vortices are random and vortices change their locations from one experiment to another. There is no link between the locations and potential heterogeneities in our system. Structural stability of the vortical structures is facilitated by alignment interactions realized through inelastic collisions (hydrodynamic, electrostatic) leading to a buildup of velocity and spatial correlations.

4. *When $\kappa=0$, I would have expected a standard flocking phase to emerge (instead of β vortices). Could the authors explain why it is absent from the phase diagram? Does this reflect the polydispersity of the curvature distribution that does not peak at 0 when the $\langle \kappa \rangle = 0$?*

Indeed, the Reviewer is correct. The $\langle \kappa \rangle = 0$ case should correspond to classical spherical Quincke rollers with a standard flocking regime. However, it is absent from our phase diagram due to unavoidable polydispersity of the experimental chiral rollers (shape, size, surface chemistry) leading to a polydispersity of curvatures. $\langle \kappa \rangle = 0$ in the case of experimental system means that there is almost equal amount of particles with negative and positive κ in the transition region. This results in the β -vortices comprised of rollers of both (heads-in and heads-out) orientations as shown in Fig 4c.

5. *At the single-colloid level, could the author provide a qualitative explanation for the chirality reversal? (a quantitative explanation would be even better)*

This is, of course, an important question. However, nontrivial. To provide an adequate answer one needs to compute z-component of the electrostatic torque acting on a pear-shaped roller by solving dynamical equation for the polarization of the shape-anisotropic particle in a liquid in the presence of a bottom electrode, and coupled to the angular momentum conservation equation. For spherical Quincke rollers this component of the electrostatic torque could be deduced analytically and is always zero due to symmetry (see Ref.27). In the case of a pear-shaped roller it is nontrivial and could be done only numerically. This by itself could be a matter of a separate research. Qualitatively such procedure should result in a non-zero z-component of the electrostatic torque acting on a pear-shaped particle that changes sign with the strength of the electric field. In the current paper, we are focusing mostly on the rich phase behavior and characterization of this new system.

6. *Fig. 1b. Do the authors plot the most probable or the average speed value. The suppression of a sharp bifurcation could well be a mere artifact caused by some polydispersity in the particle shape/chemistry. Plotting the most probable value of the propulsion speed should suppress this possible artifact.*

Fig 1b has been plotted using the average speed values shown with standard deviations. We agree with the Reviewer that some polydispersity in the system might suppress a sharp crossover. Following the suggestion, we re-plotted the Fig1b panel using most probable velocities. However, that provided almost no improvement. The results are below:

7. *Termining κ "curvature" might be misleading, $1/\kappa$ is not a length scale.*

We use κ as a normalized "curvature parameter". The way we clearly define it in the text should prevent any confusion.

8. As Fig. 1.d hardly depends on E , I feel that a 2D plot of $\kappa(E)$ could provide a clearer illustration of the change in the roller behavior.

We agree with the referee and replaced the Fig 1d panel with a 2d plot of κ versus E . We moved the old panel with data obtained at different rollers densities to the Supplementary materials.

Reply to Reviewer 3

We thank the Reviewer for a careful reading of our manuscript and finding our results novel. The Reviewer made a number of comments and questions which we fully addressed in the revised manuscript. Below are point-by-point replies to the Reviewer's comments:

1.- *First, the abstract and introduction do not give clear hints of what is done in the manuscript and have in fact very little information about. In the conclusion paragraph, it is only the first few lines that summarize the work, while the second half refers to unproven and I would say unlikely projections of their results.*

We re-wrote the abstract and modified the conclusions section to better indicate what has been done and reported in the manuscript.

2.- *In the first part of the manuscript the dynamics of individual particles is analyzed (around Fig. 1). The motion of the colloid is described as belonging to three modes attending to the orientation of the main axis with respect to the substrate plane. Figure 1c, and the sentence: "Transitions from α to γ modes with the activity also inevitably lead to a chiral states reversal of the individual rollers" indicate that the trajectory of mode β is straight, while the other two are curved, so one would be CW and the other CCW but which one is which one? And we can understand that fixing the applied field and the density these could not coexist then. Is this correct?*

The modes of rolling (α and γ) differ by particle orientation – either head-out or head-in- with respect to the center of the circle of the trajectory. However, the same mode (let's consider α mode as an example) can rotate on a trajectory both CW and CCW (still keeping orientation- head out for α). Thus, for a fixed mode of rolling (defined by the strength of the field) both chiralities CW and CCW are simultaneously present in the system as particles randomly select the direction of motion on the trajectory (CW or CCW) while preserving their mode of orientation.

For a moving roller a transitions from one mode of rolling to another (for instance, α to γ) results in change of the sign of the local curvature that would result in reversal of the winding - CW becomes CCW and vice versa. We added additional corresponding clarifications in the revised text.

3.- *I find the discussion of the MSD and MSAD shown in Figs 1e and 1f quite confusing.*

3a.-- *The text only states: "Initially the rollers move ballistically as $msd \sim \tau^2$ and then transition to a diffusive-like regime at longer times." As far as I see this is clear the case only for the trajectory of mode β , since mode α seems to remain ballistic much longer and mode γ can become even subdiffusive.*

3b.-- *The text further states: "At high activity levels, the curvature of the trajectories increases resulting in an appearance of characteristic oscillations in the mean square displacement". But the oscillations are only appearing in the mode β data, so should we understand that this mode is the only one showing rotating trajectories?*

3c.-- The MSAD data seems to agree with the previous statement since mode γ remains ballistic, while the other data clearly show the transition from diffusive to ballistic, so this would indicate rotational motion to all modes, contradicting previous statements

Indeed, we agree with the Reviewer that initial description was confusing. We modified the text to better describe the observed MSD and MSAD behavior of the rollers at different regimes shown in Fig 1e and Fig 1f. In particular, rollers in the regimes close to the transition point (blue and yellow curves in Fig.1e) have a low curvature of their trajectories (since they are close to a curvature transition region) compared to the rest of the curves shown and transition to a nearly diffusive regime at longer times as they are less localized than rollers with high curvature of the trajectories and, therefore, have higher probability for collisions with other rollers. As curvature and activity increases (red curve) characteristic oscillations of the mean square displacement emerge indicative of the circular-like, more localized motion of the rollers and as a result sub-diffusive behavior. Similar behavior is eventually observed also for α - mode rollers (purple curve) with high curvature of the trajectories, albeit at much longer time scales due to significantly lower velocity of the rollers motion at low field strengths. Similar oscillations are also reflected in the variance of the in-plane particle orientations $\langle \phi^2 \rangle$, shown in Fig.1f, where all regimes develop t^2 dependence indicative of predominantly circular-like motion of the rollers. Initial perceived deviations from t^2 behavior at times below 0.01s (there are only 2 points there) in some of the MSDA curves in Fig 1f are possibly related to the uncertainty in determination of small changes in the particle orientations at such small-time scale from the experimental data.

4.- Later in the manuscript collective dynamics is investigated and linked to the single particle properties. Although this would be a standard solid procedure to understand the system, here is confusing. In page 4, and related to date in Fig. 2e it is stated that: "the system of pear-shaped rollers exhibits a spontaneous chirality induced phase separation from initially random distribution of chiral rollers" The initially random distribution is absolutely not proved here, and also not the transition but only an already phase separated system. Looking very carefully at all the provided movies, I would say that it seems more that the rotation is more dictated by the environment, (Vicsek type of behavior) and not to the individual properties as discussed here.

Upon initial application of the energizing electric field both windings (CW and CCW) are simultaneously realized in the system at any mode of rolling (please see the reply to the question 2). It takes system a few seconds to develop collective phases (including flocs) driven by alignment interactions. In the revised manuscript we have included a discussion on the mechanisms leading to the formation of the collective phases. See also the reply on comment (ii) from Referee 1. We agree with the Reviewer that some of the behavior can be described by a Vicsek type models, we included a few recent computational references (Ref. 46-48) predicting rotating droplets and chirality induced phase separation accounted for by the Vicsek-like models for chiral swimmers.

5.- Discussing Fig. 2f the authors refer to the subdiffusive behavior shown in Fig. 1e, but as far as I understand these two sets of data correspond to different densities, so no linked or conclusion could be made. The MSD at higher densities should then be also measured, and the collective behavior at more dilute regimes discussed.

The Reviewer is correct. There is no direct relation between Fig 2f and Fig 1e as they are taken at very different particle number density. We removed incorrect reference to the panel from the revised text and modified the discussion accordingly.

6.- *The radius of the individual trajectories seem to be importantly affected by the intensity of the applied field and/or by the density, as shown in Figs 2c and 2f. I find this very interesting, but it has not been even discussed at the single particle regime level, and I think should be investigated for all range of E and ϕ values here presented.*

Indeed, the curvature of the trajectories (and as a result the persistence length of rollers) is controlled by the electric field strength. We experimentally investigated the behavior of the persistence length for different roller densities and electric field strength. The results are shown in Fig. 3d.

7.- *Is the velocity in Eq.(1) related with the ballistic velocity of Fig 1e, and the frequencies in the inset of Fig 3b are probably related with the angular velocities of Fig 1f? If there is a reason not to link these values it should be explained, otherwise a more in deep analysis should be presented. For example the frequencies are claimed to increase linearly with activity, but how linearly, quadratically, do the authors present an argument of why is this behavior expected?*

The data in Fig 1e and Fig 3b are not linked because they are obtained at very different (3 orders) particle densities, $\phi = 0.001$ and 0.108 as indicated in the Figure captions. We modified the captions to have these values immediately available to the reader.

The frequency of the oscillations in the velocity temporal correlation function increases with the activity. And we state that it increases without claiming exact functional dependence because it depends on two major factors as we show in the paper: change of the velocity with the field strength and shape dependent change of the curvature of the particle trajectory with the field that is nontrivial and depends strongly on the particle geometry and shape asymmetry. So, unfortunately there is no simple linear, quadratic etc. dependences. We added a comment in the revised text.

8.- *The persistence length L_p is here defined "as the distance particles travel where the velocity temporal correlation function C_t decays to $1/e$ ". I assume this has to be equivalent, or at least related to the more intuitive and more commonly used definition of the decay of the orientation correlation along the trajectory as most, is this the case?*

Correct, it is the same.

All movies and the trajectories shown in Figs. 3c, 3f show that the distribution of such L_p 's should be broad, since it can be very different depending on the particle relative position to the vortex center or boundary area. This is not discussed and I think it will have a big influence in the relevance of these quantities.

Indeed, the distribution of persistence lengths at high particle densities becomes broader while average persistence length increases in collective phases. We have Fig. 3d to demonstrate the changes of the average persistence length when moving from a gas to collective phases with density of rollers at fixed activity. It demonstrates that persistence length is not the universal parameter to probe the state of the system.

9.- *Are all the points in Fig 3a represented in Fig 3f? To keep the symbols then could bring some additional information.*

Correct, in Fig 3f we use/show information about all vortices shown in phase diagram Fig 3a.

10.- *I find very interesting that the nature of the vortices changes with the intensity of the applied field, but I have been quite confused about it. The reader is confronted to vortices α , γ that rotate always CCW, while the only movie of various coexisting vortices shows both CCW and CW rotations. In the end,*

I realized that Fig 4c corresponds to Fig 3a, and Fig. 4e to Fig 3d, while no general image is shown for Figs 4a, 4b, and 4d. To discuss this and show the corresponding movies, and plots of the velocity-vorticity and trajectories seems necessary to me.

Movies of all phases are supplied. Structure and orientational order of all the phases is demonstrated in Fig 4. Also Fig 3 g,h,i demonstrates experimental snapshots of the most spectacular phases: α and γ vortices and rotating flock.

11.- *I also assume now that the beta vortex is able to choose both CW and CCW, and therefore is this type (with its very limited set of possible parameters) the only one able to show the discussed coexisting vortexes of opposite rotations. In the text now, it seems that this behavior is more the rule than the exception.*

All types of vortices, α , β , γ , can chose CW or CCW winding. Rollers of both windings are always realized in the system with equal probability. Please see reply to question 2.

12.- *On the other hand, I am not sure if this contradicts the discussion at the beginning of the manuscript were the rotation direction was stated to be determined the applied field (see Fig. 1d). The rotations we supposed to have opposite direction for α and γ , while non-existing for β . It could be that the explanation for this is that the effect emerges with increasing density, but something should be said and probably characterized about it, or simply that the single particle motion need more clarification.*

Please see the reply to question 2.

13.- *The authors also claim around Fig. 2 that the discussed patterns are quite stable on time. However, there is not much information about for how much time is this checked, or if this happens for all types of patterns, or only for the two shown in Fig. 2.*

In a typical experiment, once the electric field is on, rollers self-assemble into the corresponding patterns and reaches a steady state within a few seconds. We start to capture 8-second videos to analyze the structure and dynamics of the particles 1 min after the electric field has been switched on. During a typical run time of the experiments (about 10 min), no apparent changes in the patterns has been observed. We added corresponding information in the revised text.

14.- *The spinner phase is only shown in the most diluted case. I would find interesting to see how this is for higher densities.*

The spinner phase in our parameter range can be realized at high density (0.1) at high electric field strengths (compared to a dilute case (0.001). We also added a discussion on the mechanisms of the transition to the spinner phase. See reply to question (ii) from Reviewer 1.

15.- *If I understand properly the PDF's of ψ_1 and ψ_2 shown in the manuscript and in more detail in the supplementary note correspond just to a very particular configuration, this is one realization at one particular time. This would mean that these peaks can change with time and would disappear if an averaged PDF would be shown. Would the rest of the distribution then remain stable in that case, and I would wonder about the relevance of the peaks is a general discussion of the system as the one presented in this work. More important would be to discuss the polar order shows two maximums in the vortex phase, but not the nematic order.*

The curves are average over time and multiple realizations of the system under the same conditions. Peaks are related to the nearest neighborhood of the particles and not noise related. The nature of the peaks is explained in detail in Supplementary Note 2.

The nematic order, ψ_2 (shown in dash), see Fig 4f, does show a maximum closer to $\psi = 1$ for all vortex phases.

16.- *Finally I am interested on the importance of the system size. Would it possible to make experiments with cell of different sizes ? Both smaller or larger would provide important information. In case this is not possible, a discussion of the possible outcome, and the consequent universality of the obtained phase diagrams seems relevant here.*

The cell size did not affect the emergence of the patterns (unless it is very small comparable to the vortex size 40 particle diameters). It would be also expected that with a very small system size, collective behaviors will be strongly affected by boundaries, which we wanted to avoid in this paper.

Other minor comments the authors might be interested on:

- *Fig 1e refers to solid line as having slope 1, but all data are also solid lines.*

We fixed typo. Now it reads "black solid guide line".

- *The frequently refereed purple color is a bit darker blue in my prints.*

Color deviation is common between different printers. However, we believe that it will not cause misunderstandings of our data in this particular case as other curves are significantly different.

- *Caption Fig 3a: I guess that the colors do not refer to the symbols as stated in the caption, but to the background color.*

Correct. We fixed the typo.

- *The scale bars would be more clear referring to the same length in all plots.*

We use the same scale bars when we show the whole system (Fig. 2, Fig. 4 main). When dealing with zoom-in images (Fig. 3, Fig. 4 insert), we use a smaller scale since the previous scale bar is too large.

- *Why there is no data for $\phi = 0.001$ in Fig. 3e ?*

Fig. 3e shows the sizes of vortices vs the field strength, while for samples with $\phi = 0.001$, there is no vortex.

- *Symbols are not always defined before being used. For example 'd' only appears in the method section, but is used in Fig.3*

-- *E* and *E0* are first used and a bit later only *E* is introduced.

Thanks for pointing it out. We correct those items in the revised manuscript.

REVIEWERS' COMMENTS:

Reviewer #1 (Remarks to the Author):

The authors have satisfactorily replied to all of my points and I would like to support the publication of this nice manuscript in Nature Communications.

Reviewer #2 (Remarks to the Author):

The experiments on the emergence of collective chiral dynamics are well described and characterised. Although none of the spectacular patterns reported in this manuscript is quantitatively explained, the experiments provide a comprehensive insight into the possible phases of anisotropic Quincke rollers. The authors have revised their manuscript and addressed all the questions raised by the reviewers. Some of the additions, however, damaged the manuscript more than they improved it. The main text includes erroneous statements and misinterpretations of earlier results. I therefore cannot recommend this manuscript for publication in Nature Communications.

"We bring to light new emergent dynamic phases, ranging from a gas of spinners to aster- like vortices and rotating flocks"

As pointed out in my first report only one of the various phases might be new. At the very least I would recommend the authors to revise their conclusions and accurately compare their findings to earlier experimental results. See also comments below.

"In contrast to colloidal circle swimmers based on electrophoretically driven Janus colloids in a rotational magnetic field [17] where in the regime of swimming on circular orbits the winding handedness of all particles is fixed by the chirality of the rotational field, circle rollers do not have an externally prescribed chirality and both chiral states are realized with equal probability."

This is not correct. The direction of rotation of the magnetic field does not set the handedness of the janus colloids used in [17]. This system realizes a perfect example of synthetic circle "swimmers". The realization of active colloidal particles undergoing unbiased orbital motion is not new.

"The pattern is reminiscent of self-organized vortices of both chiralities emerging in dense ensembles of collectively moving microtubules propelled by surface-bound molecular motors [25]. There, however formation of vortices was driven by a reptation-like motion of microtubule in combination with local nematic alignment interactions, while vortices in our system rely on a chiral motion of the units and velocity aligning interactions discussed below."

Again this statement is not correct. The authors of [25] provided evidence of polar interactions as in all self-propelled rod systems. I do not see any profound difference with the early results reported in [25]. I also became aware of this article by the Vlahovska group:

Tuning the Random Walk of Active Colloids: From Individual Run-and-Tumble to Dynamic Clustering, Hamid Karani, Gerardo E. Pradillo, and Petia M. Vlahovska Phys. Rev. Lett. (2019)

The authors report the formation of compact vortices in ensemble of isotropic Quincke rollers with finite persistence length. It seems that chirality is not essential to the emergence of this dynamical state.

"The polarization within the rotating flocks changes with the electric field strength. It first growth with the field strength as we move away from the phase boundary with the vortex phase and then decreases due to shrinking interaction radius of the chiral rollers"

Why does the interaction radius follow would this trend? Is it a conjecture or a result deduced from

quantitative measurements? I do not see any demonstration of possible modulation of the range of any of the interactions with the magnitude of the E field (neither experiments or theory) in this manuscript.

"The sizes of the vortices are controlled by the interaction range as demonstrated by the linear relation between the vortex sizes and the persistence length shown in Fig. 3f."

I do not understand this reasoning. Why would the persistence length be related to the interaction range? What microscopic effect could explain a change in the interaction radius? This highly non trivial (and uncommon) feature would require a demonstration.

"As a result, the strength of the alignment interactions has a non-monotonic dependence on the electric field magnitude evident from uncorrelated phases at both lower (Gas) and higher (gas of Spinners) field strengths, and collective phases (Vortices and Rotating flocks) observed in the intermediate region of the field strengths, see Fig. 3a."

Is the interaction range and magnitude supposed to depend on the magnitude of the E field? There is no demonstration of a non-monotonic behavior at the level of the microscopic interactions in this manuscript.

Reviewer #3 (Remarks to the Author):

The new version of the manuscript has improved in numerous aspects, which has certainly improved the overall clarity of the work. As I already stated in my previous report, the experimental results show very attractive and to my knowledge novel collective behavior brought by the asymmetry of the colloidal rollers.

I think the topic is timely and of great interest, the work is sound and it think it will a significant impact, such that I recommend it for publication in Nature Communications.

Before publication the authors might still want to consider the two following minor suggestions

- Regarding the inset of Fig 3b: I would represent the dependence of the oscillation frequency with the electric field. As the authors indicate, it will not be trivially linear since it depends also on the roller velocity, but could be possible to understand
- Figure 1f simultaneously considers data of different phases without distinction. I still think that denoting the points with the same symbols as in Fig 1a might be useful.

Reply to Reviewer 1

We thank the Reviewer for the positive judgement of our work.

Reply to Reviewer 2

We thank the Reviewer for a careful reading of our manuscript and valuable comments which we fully addressed in the revised manuscript. Below are point-by-point replies to the Reviewer's comments:

"We bring to light new emergent dynamic phases, ranging from a gas of spinners to aster- like vortices and rotating flocks". As pointed out in my first report only one of the various phases might be new. At the very least I would recommend the authors to revise their conclusions and accurately compare their findings to earlier experimental results.

Indeed, some of the phases can be observed in other active systems. We revised the statement. It now reads "... we reveal emergent dynamic phases, ranging from a gas of spinners to aster- like vortices and rotating flocks".

"In contrast to colloidal circle swimmers based on electrophoretically driven Janus colloids in a rotational magnetic field [17] where in the regime of swimming on circular orbits the winding handedness of all particles is fixed by the chirality of the rotational field, circle rollers do not have an externally prescribed chirality and both chiral states are realized with equal probability." This is not correct. The direction of rotation of the magnetic field does not set the handedness of the janus colloids used in [17]. This system realizes a perfect example of synthetic circle "swimmers". The realization of active colloidal particles undergoing unbiased orbital motion is not new.

The Reviewer is right that in the system described in ref. 17 both chiral states are realized. We removed the above statement.

"The pattern is reminiscent of self-organized vortices of both chiralities emerging in dense ensembles of collectively moving microtubules propelled by surface-bound molecular motors [25]. There, however formation of vortices was driven by a reptation-like motion of microtubule in combination with local nematic alinement interactions, while vortices in our system rely on a chiral motion of the units and velocity aligning interactions discussed below." Again this statement is not correct. The authors of [25] provided evidence of polar interactions as in all self-propelled rod systems. I do not see any profound difference with the early results reported in [25]. I also became aware of this article by the Vlahovska group: Tuning the Random Walk of Active Colloids: From Individual Run-and-Tumble to Dynamic Clustering, Hamid Karani, Gerardo E. Pradillo, and Petia M. Vlahovska Phys. Rev. Lett. (2019). The authors report the formation of compact vortices in ensemble of isotropic Quincke rollers with finite persistence length. It seems that chirality is not essential to the emergence of this dynamical state.

The purpose of the above sentence was to contrast the difference between local collisional nature of alignment interactions observed in dense ensembles of collectively moving microtubules propelled by surface-bound molecular motors versus long range hydrodynamic alignment interactions of the chiral rollers. We modified the above sentence to clarify our statement. Now it reads: "...There, however formation of vortices was driven by a reptation-like motion of microtubule in combination with local nematic alinement interactions due to collisions, while vortices in our system rely on long-range hydrodynamic aligning interactions discussed below".

"The polarization within the rotating flocks changes with the electric field strength. It first growth with the field strength as we move away from the phase boundary with the vortex phase and then decreases due to shrinking interaction radius of the chiral rollers". Why does the interaction radius follow would this trend? Is it a conjecture or a result deduced from quantitative measurements? I do not see any

demonstration of possible modulation of the range of any of the interactions with the magnitude of the E field (neither experiments or theory) in this manuscript.

Indeed, the terminology used (“interaction radius”) was confusing. The loss of polarization in the rotating flocks with the field strength happens due to growing localization of the chiral rollers (driven by increasing curvature of the trajectories that results in chiral rollers moving on circular trajectories with shrinking radius with the field strength). Localization leads to a decreased number of interactions with other chiral rollers. We rephrased the sentence accordingly: “... and then decreases due to growing localization of the chiral rollers that leads to a decreased number of interactions with other chiral rollers...”.

"The sizes of the vortices are controlled by the interaction range as demonstrated by the linear relation between the vortex sizes and the persistence length shown in Fig. 3f." I do not understand this reasoning. Why would the persistence length be related to the interaction range? What microscopic effect could explain a change in the interaction radius? This highly non trivial (and uncommon) feature would require a demonstration.

Similar with the previous comment we replaced the confusing terminology and rephrased the statement. “The size of the vortices depends on localization of the chiral rollers as demonstrated by the linear relation between the vortex sizes and the persistence length shown in Fig. 3f.”

"As a result, the strength of the alignment interactions has a non-monotonic dependence on the electric field magnitude evident from uncorrelated phases at both lower (Gas) and higher (gas of Spinners) field strengths, and collective phases (Vortices and Rotating flocks) observed in the intermediate region of the field strengths, see Fig. 3a." Is the interaction range and magnitude supposed to depend on the magnitude of the E field? There is no demonstration of a non-monotonic behavior at the level of the microscopic interactions in this manuscript.

Indeed, there is no direct demonstration of a non-monotonic behavior of the interaction length with the field amplitude. We rephrased the above statement to read: “As a result, the effect of the electric field on the collective behavior is non-monotonic that is evident from the presence of uncorrelated phases at both lower (Gas) and higher (gas of Spinners) field strengths, and collective phases (Vortices and Rotating flocks) observed in the intermediate region of the field strengths, see Fig.3a.”

Reply to Reviewer 3

We thank the Reviewer for the positive evaluation of our work. The Reviewer suggested minor improvements to our manuscript that we address below.

- Regarding the inset of Fig 3b: I would represent the dependence of the oscillation frequency with the electric field. As the authors indicate, it will not be trivially linear since it depends also on the roller velocity, but could be possible to understand

Following the Reviewer’s recommendation, we replaced the insert of Fig 3b with the graph of the oscillatory frequency versus electric field.

- Figure 1f simultaneously considers data of different phases without distinction. I still think that denoting the points with the same symbols as in Fig 1a might be useful.

Fig. 1f shows data only in one phase (gas) obtained at very low density $\phi = 0.001$ as indicated in the captions.